# Mitigating Early Phase Separation of Aliphatic Random Ionomers by the Hydrophobic H-Bond Acceptor Addition

**David Julius, Chunliu Fang, Liang Hong *** and **Jim Yang Lee ***

Department of Chemical and Biomolecular Engineering, National University of Singapore, 10 Kent Ridge Crescent, Singapore 117585, Singapore; david_julius@hotmail.com (D.J.); fangchunliu@nus.edu.sg (C.F.)
* Correspondence: chehongl@nus.edu.sg (L.H.); cheleejy@nus.edu.sg (J.Y.L.)

**Abstract:** This study reports a new phenomenon whereby the ionic content of a random ionomer was increased by the introduction of a hydrophobic modifier. In the current study, the ionomer synthesized from the solution polymerization of the three vinyl monomers, which are polar hydrophobic monomers acrylonitrile (AN), glycidyl methacrylate (GMA), and ionic monomer potassium 3-sulfopropyl methacrylate (SPM), encountered an early phase separation problem when the ionic content exceeded a certain threshold value. However, the addition of a strongly hydrophobic monomer, 2,2,3,3-tetrafluoropropyl methacrylate (TFPM), during the copolymerization is able to restrain this phase separation trend, consequently allowing 50% more of SPM units to be incorporated and uniformly distributed in the ionomer and achieving a random copolymer chain. The ionic clustering of the SPM units, which is the cause for the phase separation, was reduced as a result. The resulting random ionomer was demonstrated to be a superior proton conducting material over its ternary originator. This is due to the fact that TFPM possesses acidic protons, which brings about an association of TFPM with SPM and GMA via hydrogen bonding. This study could impact the synthesis of random ionomers by free radical polymerization since monitoring ionic content and improving ionic unit distribution in ionomers are issues encountered in several industries (e.g., the healthcare industry).

**Keywords:** ionomer; polymer electrolyte membrane (PEM); solution copolymerization; fluorine-containing methacrylate; H-bonding mediated copolymerization

## 1. Introduction

Random ionomers are useful for a wide range of applications. The fabrication of polymer electrolyte membranes (PEMs) for DAFCs represents one of these applications [1–4]. A relatively simple design and ease of synthesis are the advantages of random ionomers over ordered (graft, block, or star) ionomers [5,6]. However, the controllability of phase separation during membrane formation, which is needed for establishing a pervasive hydrophilic network, is more difficult for random ionomers [7,8]. The resulting membranes are therefore lower in proton conductivity than those fabricated from a more ordered ionomer structure (such as Nafion®). Since the proton conductivity of ionomers increases generally with the ionic content (i.e., the sulfonic acid group), the synthesis of random ionomers with a high ionic content is an obvious solution to attain a high proton conductivity for PEMs constructed from random ionomers [9–11].

Solution polymerization and emulsion polymerization are two common methods to prepare random ionomers [12–15]. However, the synthesis of random ionomers in the solution polymerization system often encounters two technical problems: (i) the lack of a common solvent for both the hydrophobic monomer(s) (the main components for maintaining mechanical property) and the ionic monomer (the minority component for supporting ionic conduction); and (ii) early phase separation during polymerization caused by the non-uniform distribution of the ionic units in the random copolymer chains and the

ionic clustering of the ionic segments [16,17]. Polymerization in an oil-in-water emulsion has been used to distribute the dissolved ionic monomers (in water) more uniformly in the growing hydrophobic chains through additions at the micellar interface [18]. However, the ionic content of the copolymer that can be incorporated by the emulsion polymerization is quite limited. Consequently, random ionomers with high ionic contents are now mostly prepared by pre-synthesizing a hydrophobic copolymer with functional groups on side chains that can be converted into ionic groups separately [14,16]. The cost of this approach, however, can be rather high due to involving a more complex synthesis process.

A one-pot solution polymerization to obtain random ionomers with a sufficiently high ionic content is clearly preferable if the critical issue of phase separation during copolymerization can be satisfactorily overcome. In this regard, the work of Kim and coworkers is noteworthy [19]. These authors used a specifically synthesized reactive compatibilizer (modifier), i.e., the amphiphilic urethane acrylate non-ionomer (UAN) precursor, to successfully avert phase separation in the solution polymerization of PS-*co*-PSSA aliphatic random ionomers. A relatively high ionic content (up to 20 wt.%) could be incorporated in an ionomer if a large amount of UAN was used. Other than this approach, the authors of this paper are not aware of other attempts at using additives in solution copolymerization to address the phase separation problem.

The present report describes the synthesis of a series of aliphatic random ionomers containing co-monomeric units of acrylonitrile (AN) and methacrylates (GMA, SPM, and TFPM). AN is the main constituent for forming a strong ionomer backbone owing to the dipole-dipole interaction between the AN units. Together with the methacrylates, the AN units provide structural stability for the fabricated membranes. The design also leverages on the alcohol resistance of AN and methacrylates (GMA and TFPM) to minimize the alcohol crossover problem and the ionizability of the SPM units to support proton transport. The three co-monomeric (AN, GMA, SPM) system, however, was unable to produce high ionic content random ionomers due to early phase separation, resulting in the formation of flocculant. The precipitation of the ternary random ionomers, P(AN-*co*-GMA-*co*-SPM), occurred during the free radical solution polymerization at relatively low SPM loadings (<12 mol%). However, the SPM content in the ionomer could be increased by adding an appropriate amount of the strongly hydrophobic TFPM to the polymerization solution, forming the quaternary random ionomer P(AN-*co*-GMA-*co*-SPM-*co*-TFPM) as a result. The solution copolymerization in this case could then proceed without precipitation driven by early phase separation. The assimilation of TFPM into the growing free radical chains during copolymerization could improve the distribution of the ionic SPM units in the copolymer backbone, consequently reducing the tendency of precipitation driven by aggregation of the ionic segments. The inclination of the ionic aggregation escalates when the ionic segments become longer. This phenomenon of increasing the ionic content of a random ionomer via hydrophobic modifier insertion is counter-intuitive and has not been observed before. The effect is deemed to be a result of the association of the acidic methylene ($CH_2$) and methine (CH) protons in TFPM with the sulfonate of SPM via hydrogen bonding. The findings here could therefore provide a strategy for the design of random-ionomer membranes with high ionic contents.

## 2. Experimental

### 2.1. Materials

Potassium 3-sulfopropyl methacrylate (SPM, 98%), acrylonitrile (AN, ≥99%), glycidyl methacrylate (GMA, 97%), 2,2,3,3-tetrafluoropropyl methacrylate (TFPM, 99%), 1-hydroxycyclohexyl phenyl ketone (99%), N,N-dimethylformamide (DMF, HPLC-grade), ethylene diamine (EDA, 98%), ethanol (analytical grade) and sodium poly(styrene sulfonate) (PSSNa) ($\overline{M}_w$ = 70,000) were purchased from Sigma-Aldrich (St. Louis, MO, USA). The GMA monomer was further purified by passing it through a hydroquinone-based inhibitor removal column (Sigma-Aldrich). All other chemicals were used as received. Water was deionized through a Milipore Milli-Q Water system (Sigma-Aldrich).

### 2.2. Synthesis of Random Ionomers

The first series of random ionomers, namely ternary P(AN-*co*-GMA-*co*-SPM), was synthesized as follows. AN (3.2–3.5 mmol; 64–70 mol%), GMA (1 mmol; 20 mol%), SPM (0.5–0.8 mmol; 10–16 mol%) and DMF (10 mL) were simultaneously introduced to a glass tube. 1-Hydroxycyclohexyl phenyl ketone (5 wt.% of the monomers total), used as the initiator of free-radical polymerization, was added next. After several minutes of vigorous stirring, the mixture was deaerated with argon for 30 min, and the glass tube was sealed. The sealed tube was transferred to a UV-reactor (Spectrolinker$^{TM}$ XL-1500 UV cross-linker with an intensity of 5500 W/cm$^2$, Spectro-UV, Farmingdale, NY, USA) and irradiated for 6 h to complete the polymerization reaction shown in Figure 1. The polymer solution was then transferred to a petri dish and heated at 80 °C for 24 h in a vacuum oven to form a solid product. The second ionomer series, quaternary P(AN-*co*-GMA-*co*-SPM-*co*-TFPM), was synthesized in a similar fashion in the presence of TFPM (0.25–1.0 mmol; 5–20 mol%) in the initial monomer feed. A higher mole fraction of the SPM monomer (0.5–0.9 mmol; 10–18 mol%) could be used in this case without early phase separation during the solution polymerization.

**Figure 1.** Synthesis of random ionomers via free-radical solution polymerization: P(AN-*co*-GMA-*co*-SPM) (**A**) and P(AN-*co*-GMA-*co*-SPM-*co*-TFPM) (**B**).

The random ionomers are denoted by SX-Y, where X and Y are the mol% of TFPM and SPM used in the starting mixture. The starting mixture also contained 20 mol% of GMA with AN making up the balance. Thus, S0-12 refers to the ternary ionomer synthesized with a starting mixture of 12 mol% SPM, 20 mol% GMA, and 68 mol% AN. Similarly, S20-14 is the quaternary ionomer synthesized from 20 mol% TFPM, 14 mol% SPM, 20 mol% GMA, and 46 mol% AN.

### 2.3. Fabrication of Random-Ionomer Membranes

For the membrane fabrication, the ionomer solution (0.1 wt.% in DMF) was cross-linked with EDA (0.5 mmol) by the reaction between the epoxide group of GMA and the amine group of EDA via nucleophilic addition reaction. The dose of EDA could theoretically crosslink about 10% by mole of the pendant GMA units. Cross-linking began with mixing the initial reaction at room temperature for 4 h under vigorous stirring. The solution was then cast on a Teflon dish and cured at 80 °C for 24 h. A uniformly transparent membrane was formed, which could be easily separated from the Teflon surface. The membrane was

then equilibrated in 1.0 M sulfuric acid ($H_2SO_4$) for 24 h at room temperature to convert the SPM units from the salt form ($K^+$) to the acid form ($H^+$).

### 2.4. Characterization

#### 2.4.1. Proton Conductivity

The proton conductivities of the cast membranes were measured by the standard four-probe method on an Autolab PGSTAT30 potentiostat/galvanostat equipped with an electrochemical-impedance analyzer (Artisan Technology Group, Champaign, IL, USA). The frequency range of 1 MHz to 50 Hz was used. All samples (as 1 cm × 3 cm strips) were equilibrated in deionized water for 24 h prior to the measurements. The membrane resistance was determined from the Nyquist plot of the complex impedance ($Z''$ vs. $Z'$) using a method described elsewhere [20]. Proton conductivity was then calculated as $\sigma = \frac{L}{RS}$, where L and S are the distance between the two electrodes (fixed at 1 cm) and the cross-sectional area of the membrane, respectively. The proton conductivity of a commercial Nafion® 117 membrane measured this way was between 0.05 and 0.07 S/cm. Its good agreement with the literature values is a validation of the measurement method [21,22].

#### 2.4.2. Reduced Viscosity and Zeta Potential

The reduced viscosities of random ionomer solutions in the two different solvent systems were measured by an Ostwald capillary viscometer thermostated at 23 ± 0.1 °C. Two solvents are pure DMF and a mixture of DMF and ethanol ($v/v$ = 1). They were used to successively dilute a stock solution of random ionomer prepared in DMF (50 mg/mL) to several concentrations in the range of 0.1 g/dL to 0.01 g/dL. The detailed procedure can be found in the literature [23]. For the zeta potential measurement, the selected ionomers were dissolved in a mixture of ethanol and water ($v/v$ = variable) by using an analyzer (Zetasizer Nano ZS, Malvern Instruments Ltd., Worcestershire, UK).

#### 2.4.3. Laser Light Scattering (LLS)

Static light scattering (SLS) measurements were performed on a Brookhaven BI-200SM (Brookhaven Instruments Corporation, Holtsville, NY, USA) using a 30 mW He-Ne laser with wavelength of 633 nm. The pinhole wheel opening was 2 mm. The measurement angles were varied from 30° to 150° in steps of 15°. The weight average molecular weight ($\overline{M}_w$) of the random ionomers was determined by a software based on the Brookhaven-Zimm plot. Dilute solutions (1–10 mg/mL) of the ionomer samples (S0-10, S5-10, S10-10, and S20-10) in DMF after passing through a 0.45 μm Millipore Millex® HN filter (Merck KGaA, Darmstadt, Germany) were used for the measurements. A dilute solution of sodium poly(styrene sulfonate) in water with a $\overline{M}_w$ of ~70,000 was used as the control.

Dynamic light scattering (DLS) measurements on a Zetasizer NanoZS (Malvern) were used to determine the size distribution of the micelles formed by adding the ionomer solution in DMF to water. The solutions of ionomers S0-10 and S20-10 in DMF were used as representative samples. The two ionomer solutions were used to prepare stable colloidal dispersions by adding them dropwise into an excess pool of water with shaking. Each solution was then separately dialyzed in deionized water for 24 h to remove the DMF [24]. The aqueous colloidal dispersions prepared as such were kept sealed at room temperature before the SLS measurements.

#### 2.4.4. Transmission Electron Microscopy (TEM)

A JEOL 2100 microscope (Pleasanton, CA, USA) was used to obtain TEM images of the ionomer particles in the colloidal dispersions prepared above. TEM samples were prepared by dispensing a drop of the ionomer solution in water onto a TEM grid.

#### 2.4.5. Ultraviolet-Visible Spectroscopy

The structures of the random ternary and quaternary ionomers, represented by S0-10 and S20-10, respectively, were examined by UV-Vis spectroscopy and $^1$H-NMR spectroscopy.

The UV-Vis spectra of the monomer mixtures in the wavelength range of 190 to 400 nm before and after polymerization were recorded on a Shimadzu UV-2450 spectrophotometer (Kyoto, Japan), for which DMF was used as solvent. [1]H NMR spectroscopy of two selected solid ionomer samples was performed on a 500-MHz Bruker Avance 500 spectrophotometer (Billerica, MA, USA), for which DMF-d7 was used as solvent.

## 3. Results and Discussion

### 3.1. Effects of Hydrophobic Modifier Addition on Phase Separation during Copolymerization

During the synthesis of the ternary random ionomers in DMF, phase separation, which was indicated when the polymerization solution was turning turbid, occurred when the SPM content in the starting mixture was higher than 12 mol%. Depending on the starting mixture composition, the phase separation could occur very early or much later during the copolymerization (Table 1).

**Table 1.** Phase separation behavior in the copolymerization of SX-Y random ionomers and resulting SX-Y membranes.

| Copolymer SX-Y * | SPM/AN/ TFPM (mol%) | After Initiation of Polymerization | After 6 h of Reaction |
|---|---|---|---|
| S0-10 | 10/70/0 | H | H |
| S0-12 | 12/68/0 | H | T |
| S0-14 | 14/66/0 | T | T |
| S0-16 | 16/64/0 | T | T |
| S5-10 | 10/65/5 | H | H |
| S5-12 | 12/63/5 | H | H |
| S5-14 | 14/61/5 | T | T |
| S5-16 | 16/59/5 | T | T |
| S10-10 | 10/60/10 | H | H |
| S10-12 | 12/58/10 | H | H |
| S10-14 | 14/56/10 | H | H |
| S10-16 | 16/54/10 | T | T |
| S20-10 | 10/50/20 | H | H |
| S20-12 | 12/48/20 | H | H |
| S20-14 | 14/46/20 | H | H |
| S20-16 | 16/44/20 | H | H |
| S20-18 | 18/42/20 | H | H |

H, homogeneous; T, turbid; * The mole fraction of GMA was fixed at 20 mol% in all batches of the monomer feed.

Such phase instability was most likely caused by the intra and inter chains clustering of the ionic SPM segments in the copolymer chains since the SPM segments with growing become more incompatible with other monomeric units and with the solvent. No phase separation occurs for the co-polymerizations without SPM. Furthermore, self-aggregation is a common observation in the polymerization of SPM [25]. Interestingly, in the synthesis of the quaternary ionomer series in DMF, phase separation did not occur until the SPM content in the starting mixture was higher than 18 mol%. This finding led us to infer a benevolent role of the hydrophobic TFPM monomer. We hypothesize that the presence of TFPM rendered a more uniform distribution of the shorter ionic SPM segments in the copolymer backbone than the case without TFPM. The more uniform distribution of the SPM segments reduced the flocculation tendency of SPM segments, and subsequently a higher solubility of the ionomer in the reaction medium was possible during polymerization.

The above polymerization systems are conceptually similar to a binary copolymerization system consisting of an AN type monomer and a methacrylate (RMA) [$CH_2$=$C(CH_3)OCOR$] type monomer. GMA, SPM, and TFPM are RMA type with similar monomer reactivity ratios [26–28]. The relative reactivity ratios [29] measured from the copolymerization of AN (1) and RMA (2) were $r_1 \cong 0.14$ and $r_2 \cong 1.20$, respectively, suggesting that during the growth of free radical chains, RMA radicals prefer to add a monomer of their own

type, whereas the AN radicals prefer an RMA monomer. This kinetic preference would therefore result in longer RMA segments being formed very rapidly in copolymerization by the free radical mechanism resulting in a lower mole fraction of AN units in the copolymer chains. Based on this AN-RMA copolymerization model, it may be deduced that the above three RMA monomers in the present copolymerization system would join the growing free radical chains faster than AN would, provided that steric hindrance and the polarity of the –OR groups did not weigh in heavily on the copolymerization rate. This would be the case when the mol% of TFPM was either zero or very low. Under such conditions, there would be extended RMA sequences (GMA and SPM), respectively, in the copolymer chains formed, resulting in long SPM segments and hence poor solubility in DMF. On the contrary, for radicals with TFPM units at the chain growing ends, the SPM monomer addition was facilitated due to hydrogen bonding that associates the TFPM unit with SPM and with GMA, respectively, as illustrated in the drawing (inset of Figure 2). Therefore, untrivial parts of SPM and GMA units could be brought into the growing radical chains through the bridging role of H-bonding. Such association, on the other hand, lowers down the reactivity of RMA owing to steric hindrance and would encourage inclusion of AN units among them. The consequence of this H-bonding mediation is that the SPM units became more dispersed in the copolymer chains and a higher mole fraction of AN compared with the case without TFPM, thereby deferring the precipitation of the ionomer in the polymerization medium caused by the clustering of the ionic SPM segments.

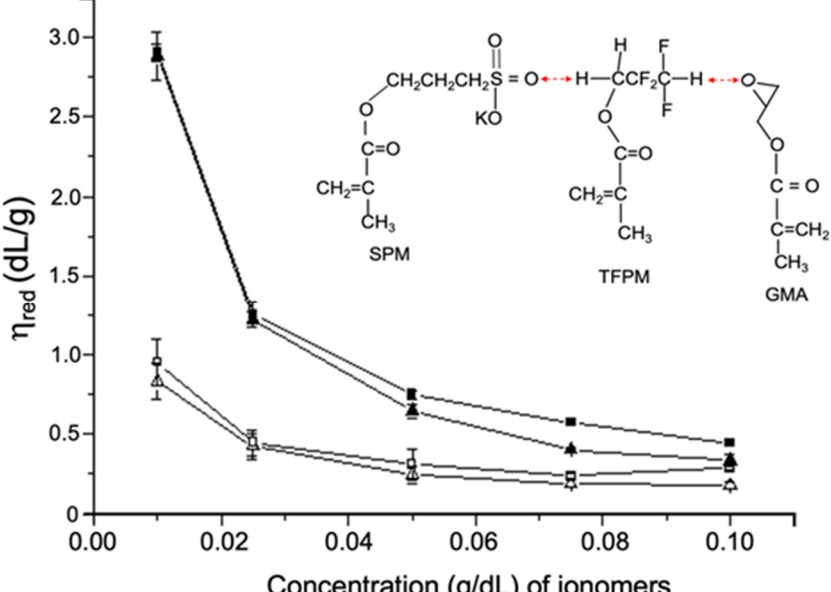

**Figure 2.** Reduced viscosity ($\eta_{red}$) as a function of the random ionomers concentration at 22·°C: ionomer S0-10 in DMF ($\Delta$) and in a DMF/ethanol mixture ($v/v$ = 1) (▲); ionomer S20-10 in DMF ($\square$) and in a DMF/ethanol mixture ($v/v$ = 1) (■). The inset shows the copolymerization mediating role of TFPM.

*3.2. Influence of Co-Monomer Distribution on Ionomer Solvation in Aqueous Medium*

The structural change from a ternary random ionomer to a quaternary random ionomer due to the presence of TFPM units in the copolymer leads to changes in solvation behavior in solvents. In this study, measurements of solution properties, such as the reduced viscosity, light scattering properties, and zeta potentials, all support the above observation, which was also supplemented by TEM examinations of the ionomer particles.

Figure 2 shows the reduced specific viscosity ($\eta_{red}$)—concentration plots of ternary S0-10 and quaternary S20-10 ionomers in the two different solvents (DMF and a 1:1 ($v/v$) DMF/ethanol mixture). DMF was found to be a good solvent for all of the monomers except for SPM. It is nevertheless a weaker solvent than DMF/ethanol for the ionomers. The

measured viscous behavior of the ionomers in the two solvents is typical of the response of the ionomers to the changes in solvent polarity and concentration [30,31].

In general, reduced viscosity ($\eta_{red}$), a physical quantity using the viscosity of solvent as reference, is a measure of the expandability of the polymer chains in a dilute solution. A good solvent therefore exhibits a higher reduced viscosity. The higher viscosities of the two ionomers in the 1:1 DMF/ethanol solvent (Figure 2), therefore, suggests a greater expansion of the ionomer chains in this binary solvent mixture. This indicates stronger ionomer-solvent interactions, which could be caused by the affinity of ethanol with SPM segments and the full solvation of AN and MMA segments by DMF, respectively (namely the cooperative solvation). Such expansion extents would not be possible with ethanol or DMF alone. Van der Waals forces and other intermolecular forces (e.g., hydrogen bonding or donor-acceptor complexation) sustaining solvation cause a negative mixing enthalpy and a positive mixing entropy. As regards the stiff decrease in the reduced viscosity with changes in concentration, this could be understood in terms of chain entanglement: at high concentrations when coils of ionomer chains were closer to one another, the increasingly stronger ionic interactions between the ionic segments led to more chain entanglement. The increase in chain entanglement lowers the contact between the ionomer and the solvent molecules, and as a result the reduced viscosity is lower [32,33].

More informatively, S20-10 ionomers exhibit a slightly greater $\eta_{red}$ than S0-10 and are less sensitive to the increase in concentration in both solvents (the S20-10 trend lines are above the corresponding S0-10 trend lines). In the binary solvent, this difference appears in the concentration range of 0.03 g/dL and above, but in DMF, the gap exists in all concentrations. This implies a higher expansivity of quaternary ionomer than its ternary counterpart. Since both ionomers have the same SPM contents, S20-10 experiences less contraction, which must have been caused by more broadly SPM distribution in its copolymer chains. Hence, this obstructs approaching of the SPM segments in the ionomer chains and could therefore more effectively oppose the proclivity for chain entanglement. Moreover, the effect of chain structure (composition and monomer distribution) on chain expansion in the very dilute environment is more obvious in a weaker solvent (e.g., DMF) than in a stronger solvent (e.g., DMF-Ethanol).

Alternatively, since the solvation of random ionomers in a polar solvent depends on the solvent-accessibility to the different types of segments, this would thereby impact the ionization of the ionic units. Indeed, this was demonstrated in the present work. Both S0-10 and S20-10 ionomers were negatively charged due to the cationic exchange nature of the SPM units. They indeed display different magnitudes of negative zeta potential with a variation of concentrations in the solvent (Figure 3). Despite containing the same SPM contents in both ionomers, this observation is attributed to the different extents of exposure to water of the SPM units in the solvated copolymer coils. Zeta potential measurements complements the results of the reduced viscosity measurements.

Figure 3 shows the zeta potential changes with the lowering of the polarity of the ethanol/water solvent. The zeta potential of the S0-10 ionomer in pure water was much more negative than that of the S20-10 ionomer. This implies that S0-10 ionomer attains the maximum chain expansion in pure water due to longer SPM segments. Consequently, the hydrophilicity of SPM was not sterically shielded by the neighboring hydrophobic segments. With the increase in ethanol contraction, ionization extent of SPM is constrained and hence the ionomer carries less of a negative charge. Additionally, S0-10 bears a near zero zeta potential in pure ethanol, implying that the SPM segments are wrapped up by the neighboring hydrophobic segments. In contrast, S20-10 possesses a rather stable negative charge level with the increase in ethanol concentration. This means that SPM and TFPM units as well as GMA units in the copolymer chain are in proximity, leading to a rather limited and stable exposure of SPM units to water (and therefore similar ionization degrees of the units). A uniform distribution of SPM units, due to the H-bonding assisted chain growth mechanism, engenders the regio-partitioning by the proximate hydrophobic segments.

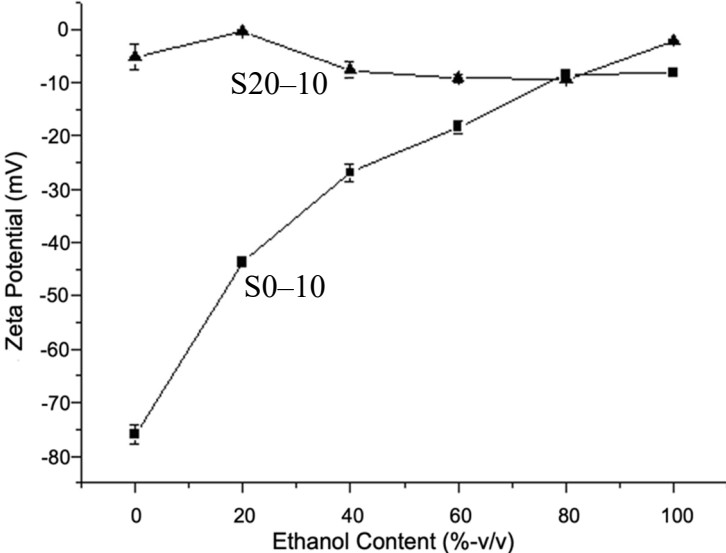

**Figure 3.** Zeta potentials of 1 wt.% S0-10 (■) and S20-10 (▲) ionomers in an ethanol/$H_2O$ mixture.

Dynamic light scattering (DLS) measurements also support this inference: a larger hydrodynamic volume was measured for the S0-10 ionomer in pure water due to a higher water-swelling extent of the SPM units (Figure 4). It is rational that S0-10 particles have a hydrogel shell on each particle, but S20-10 particles are shrunk due to the regio-partitioning effect (as proposed above). Correspondingly, TEM examination of the size and size distribution of the S-10 and S-20 ionomer particles also led to the same conclusion (Figure 5). The two TEM images provide discernable observation to support the Gaussian distribution displayed in Figure 4. Measurements of the weight-averaged molecular weights ($\overline{M}_w$) of S0-10 and S20-10 ionomers by SLS in DMF (Table 2) also show an increase in $\overline{M}_w$ with the increase in the mole fraction (*x*) of TFPM in the polymerization starting mixture. We suggest that this happens since TFPM, once joining the radical chain, stabilizes the radical formed due to the electron-withdrawing effect and steric hindrance effect of the TFP group. This therefore favors the growth of radical chains.

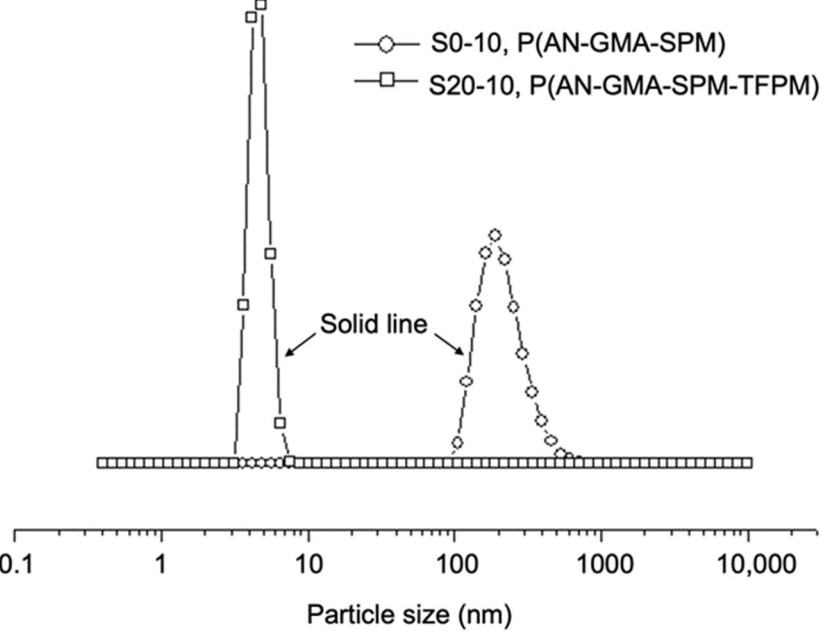

**Figure 4.** Size distribution of random ionomer aggregates formed in aqueous solution. The solid line is the Gaussian fit.

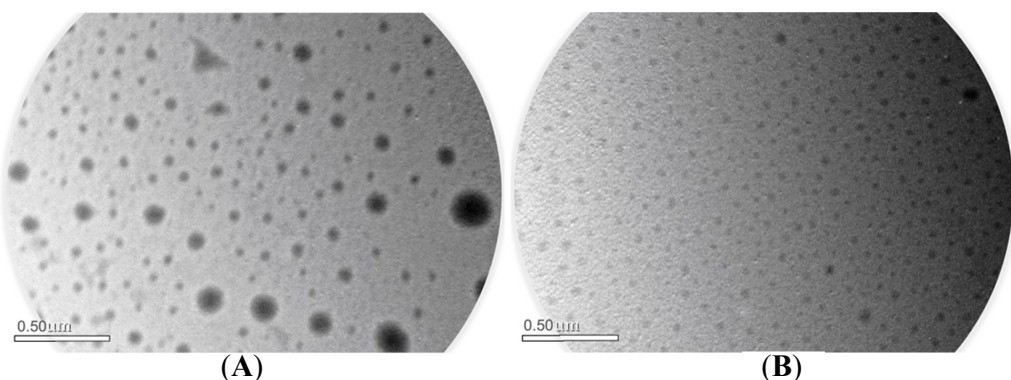

(**A**)                                                    (**B**)

**Figure 5.** TEM images (50,000× magnification) of ternary random ionomer (S0-10) aggregates (**A**) and quaternary random ionomer (S20-10) aggregates (**B**) in aqueous solution after dialysis. The initial ionomer concentration was 1 wt.%. The error bars in both images represent a 0.5 μm scale.

**Table 2.** Weight-averaged molecular weights ($\overline{M}_w$) of random ionomers from SLS measurements.

| Ionomer | SPM Content (mol%) | TFPM Content (mol%) | $\overline{M}_w \times 10^{-3}$ (g/mol) |
|---|---|---|---|
| S0-10 | 10 | 0 | 930 |
| S5-10 | 10 | 5 | 1580 |
| S10-10 | 10 | 10 | 1620 |
| S20-10 | 10 | 20 | 2300 |
| Poly(Styrene Sulfonate) *, Mw ~ 70,000 | - | - | |

* This polymer was used to calibrate the measuring condition.

The longer life of the radicals therefore led to a greater $\overline{M}_w$. With the exception of this factor, the quaternary ionomer gains a slightly larger solvation size than the ternary ionomer in DMF (Figure 2). This favors a higher reading of $\overline{M}_w$ from SLS. On the other hand, uniformly dispersed TFPM units in the quaternay ionomer could more strongly and symmetrically scatter incident light, affecting the determination of $\overline{M}_w$. As AN is the dominant constituent in the copolymer, the uniform distribution of TFPM in the quaternary copolymer chain must have a sizable impact on $\overline{M}_w$. This aspect is validated below by UV-Vis analysis.

*3.3. Additional Characterizations of the Ternary and Quaternary Ionomer Chains*

Continuing the correlation of polymer-solvent interactions with the characterization of polymer chain structures, the content and distribution of AN unit in both ternary and quaternary ionomers could also be scrutinized by their different UV-Vis spectroscopic behaviors when they are swelled by a solvent such as DMF. The inclusion of TFPM monomer, bearing a strong electron-withdrawing group ($-OCH_2CF_2CHF_2$), could behave like AN owing to the inductive effect, taking in more AN and making AN more uniformly distributed. This change causes increases in dipoles as well as spreading of monomeric units in the copolymer chains formed [34,35]. UV-Vis spectroscopic and $^1$H-NMR spectroscopic measurements provide the supporting evidence for the hypothesis the role of the addition of TFPM.

The UV-Vis spectra of AN and resulting ionomers shown in Figures 6–8 share several common features, namely a strong absorption band at 280 nm and a feeble shoulder (for monomer) at around 330 nm caused by the conjugation between vinyl and nitrile groups. It was found from separate experiments that the homo-polymerization of AN or its copolymerization with the other monomers does not bring about a shift of the maximum absorption wavelength. On the contrary, an increase in the extent of homo-polymerization broadens the absorbance band and increases its intensity. The band width could be related with the average numbers of the aligned C≡N dipoles (i.e., alignment extent), as illustrated in the inset of Figure 6. On the other hand, the band intensity could be attributed to

the number of AN association groups, in which AN monomer or monomeric units are associated together with various spatial arrangements. Such groups exist in the monomer feed as well as in the copolymer formed. The change in absorbance intensity and it peak width with the formation of homo-PAN, which has also been observed before [36], means a significant increase in alignment and random association of C≡N dipoles.

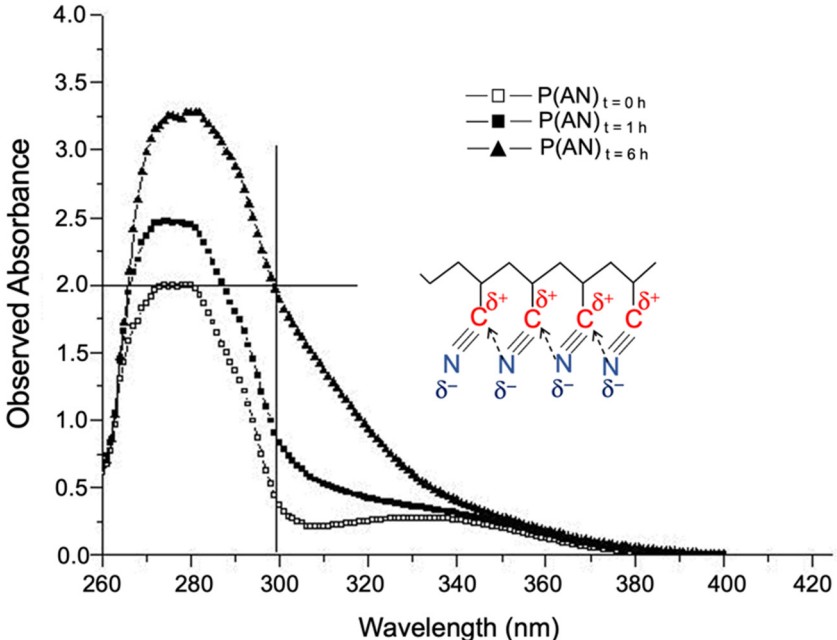

**Figure 6.** UV-Vis spectra of AN monomer in DMF before polymerization (□), after 1 h of polymerization (■), after 6 h of polymerization (▲).

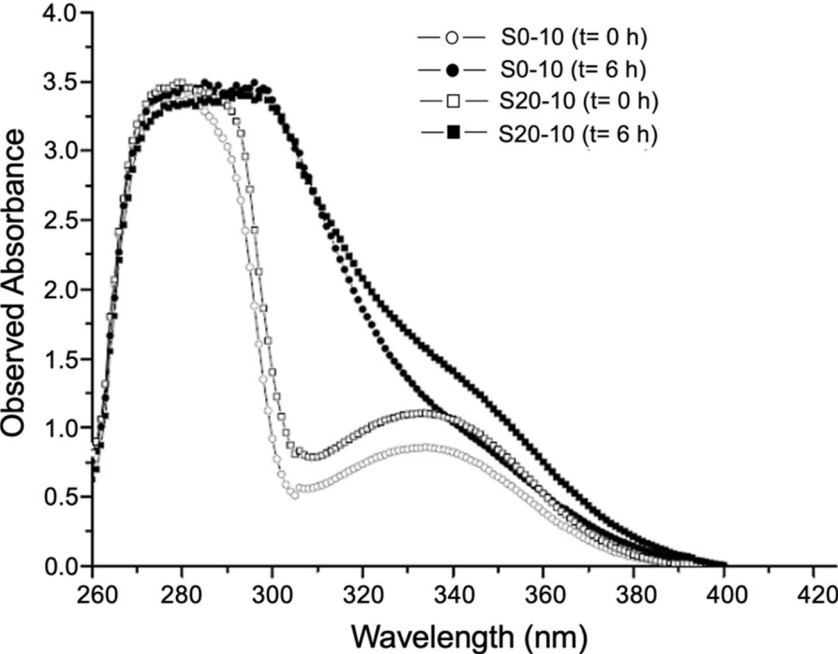

**Figure 7.** UV-Vis spectra (at the $\lambda_{max}$ of AN) of the two ionomers before and after 6 h of copolymerization: the monomer mixture S0-10 (○) and its partially polymerized product (●); the monomer mixture S20-10 (□) and its partially polymerized product (■).

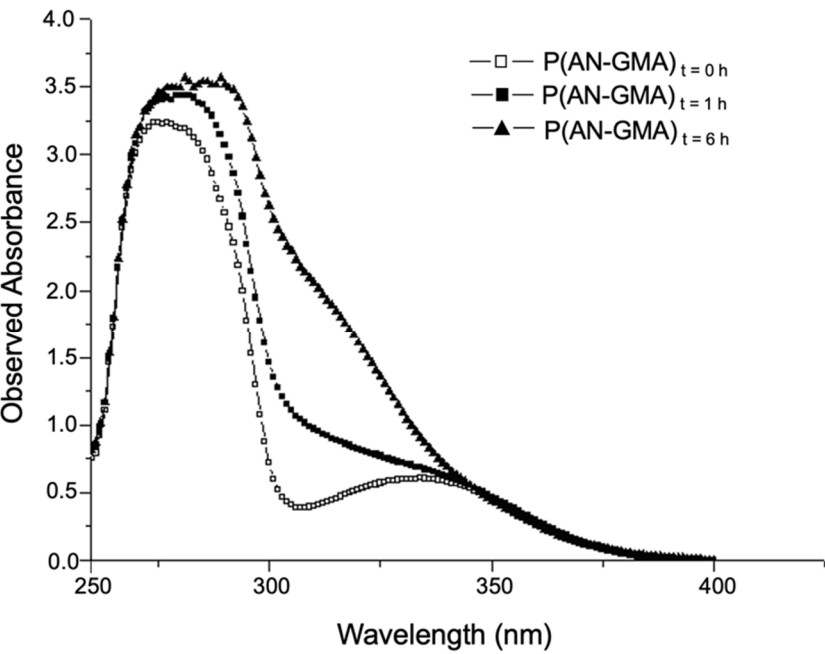

**Figure 8.** UV-Vis spectra of AN-GMA monomer mixture (with equimolar ratio) in DMF before polymerization (□), after 1 h of polymerization (■), and after 6 h of polymerization (▲).

Two copolymerization systems (i.e., S0-10 and S20-10) were selected to examine their UV-Vis spectra (Figure 7). The ~280 nm absorption is similar at the start of polymerization despite the fact that the quaternary feed presents a slightly broader band. In both systems, the absorption bands were broadened after polymerization for 6 h, indicating the formation of AN segments in the two ionomers. Different from the homo-polymerization of AN, the ~280 nm intensity is practically unchanged. This is indicative of a negligible change in the number of AN association groups after polymerization, as defined above. On the other hand, the copolymerization increases the average alignment extent of AN dipoles, leading to a broader absorption band. In regard to the comparison of UV-Vis spectra of S0-10 and S20-10, the latter presents a slightly broader absorption band than the former. This suggests the formation of a somewhat greater extent of dipole alignment in ionomer S20-10 [31]. In conclusion, AN is easier to join the radical growing chains with the assistance of TEPM, resulting in a larger number of aligned C≡N dipoles distributed among the other monomer segments of S20-10. As a control system, the copolymerization of AN and GMA exhibits the similar phenomenon (Figure 8). It suggests that more AN segments are formed with the extension of polymerization, which favors the dipole alignment in individual segments and hence broadening of the adsorption band.

In the meantime, $^{1}$H-NMR analysis of the above two ionomers revealed that the presence of TFPM in the copolymer backbone also caused down-field chemical shifts of the methylene moieties in the alkoxyl groups (-$CH_2$-O-) (marked by c, h and o) of the three RMA-type monomeric units (Figure 9). In the ternary ionomer, the methylene groups (marked by c and h) underwent peak splitting (into three peaks) due to the magnetic coupling of neighboring –$CH_2$- and >CH- groups (Figure 9A). The assimilation of TFPM segments into the ionomer causes more complicated coupling due to the introduction of $^{19}$F-$^{1}$H coupling in -$CF_2$-$CH_2$ (labeled by o). This structural change is reflected by the broadening of these three NMR peaks (Figure 9B). More importantly, the down-field chemical shifts of these three peaks suggest a reduction in the electron density surrounding the methylene moieties (-$CH_2$-) in the alkoxyl groups. This could be due to improved mixing of the AN segments with the three RMA-type segments in the ionomer chains, causing the decrease in electron density in the backbone of individual copolymer chains due to the fact that the strong electron withdrawing C≡N groups are dispersed in each quaternary ionomer chain. In short, the $^{1}$H-NMR analysis supports the occurrence that

the participation TFPM in the copolymer chains also restricts the length of AN segments without reducing the mole fraction of AN in the copolymer chains.

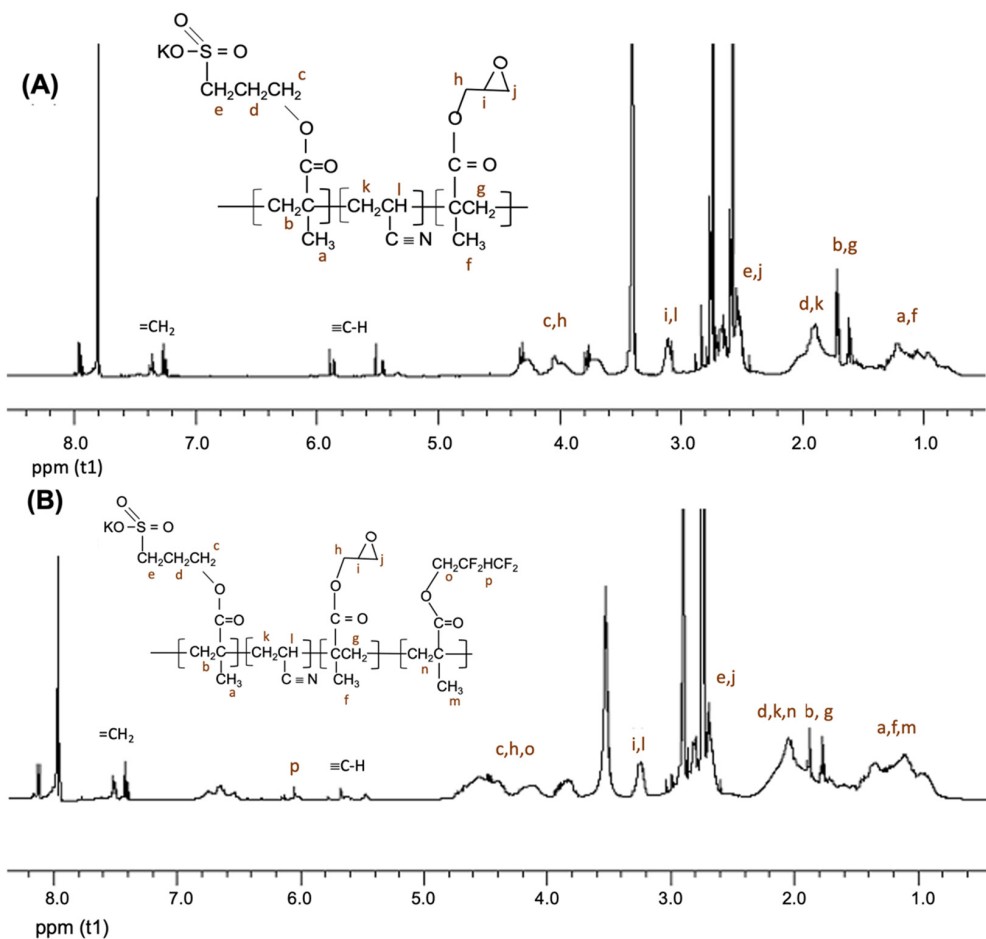

**Figure 9.** 1H NMR spectra of ionomers formed from the S0-10 (**A**) and S20-10 (**B**) systems after UV-irradiated polymerization for 6 h.

### 3.4. Effect of the TFPM Units on the Proton Conductivity

The membranes for assessing proton conductivity were prepared by solution casting technique, which has a crosslink matrix containing not only sulfonate group but also other hydrophilic pendant groups (e.g., -NH- and -OH). However, the mol% of SPM and TFPM strongly impact the proton transport in the matrix (Figures 10 and 11). It was observed during the screening of ionomer composition that no freestanding membrane could be formed with SPM content higher than 10 mol%. As shown in Figure 3, longer and separate SPM segments in the ternary ionomer make the crosslinking ineffective to restrict water swelling degree. On the other hand, low SPM contents (<10 mol%) resulted in very low proton conductivities. In samples with no or a very low TFPM content, the mol% SPM that could be incorporated into the ionomer was rather limited (Table 1). Membranes of SX-10 ionomers were examined to clarify the effect of TFPM on proton conductivity. As shown in Figure 10, the variation of mole fraction of TFPM (X = 0 to 20 mol%) shows only a minor effect on the proton conductivity. However, the water-uptake decreases with increasing TFPM content from 0 to 10. The hydrophobicity of TFPM clearly prevails in these cases over the improved distribution of SPM units since the SPM units are adequately partitioned by TFPM since X = 10 mol%. Proton transport is not enhanced as a result.

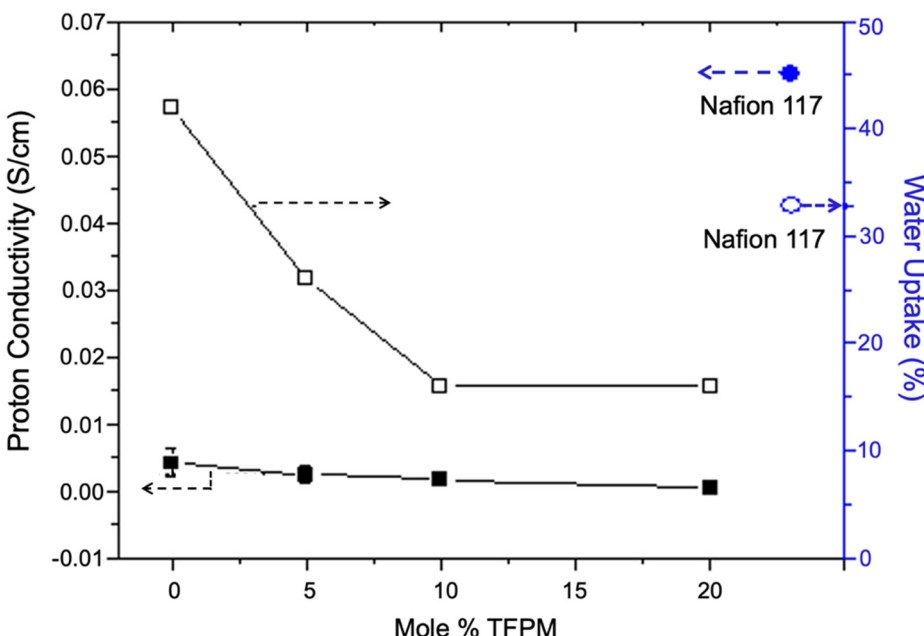

**Figure 10.** The effect of hydrophobic TFPM modification on the SX-10 proton conductivities at room temperature. The conductivities of ternary S0-10 membrane and the Nafion® 117 membrane are included for comparison.

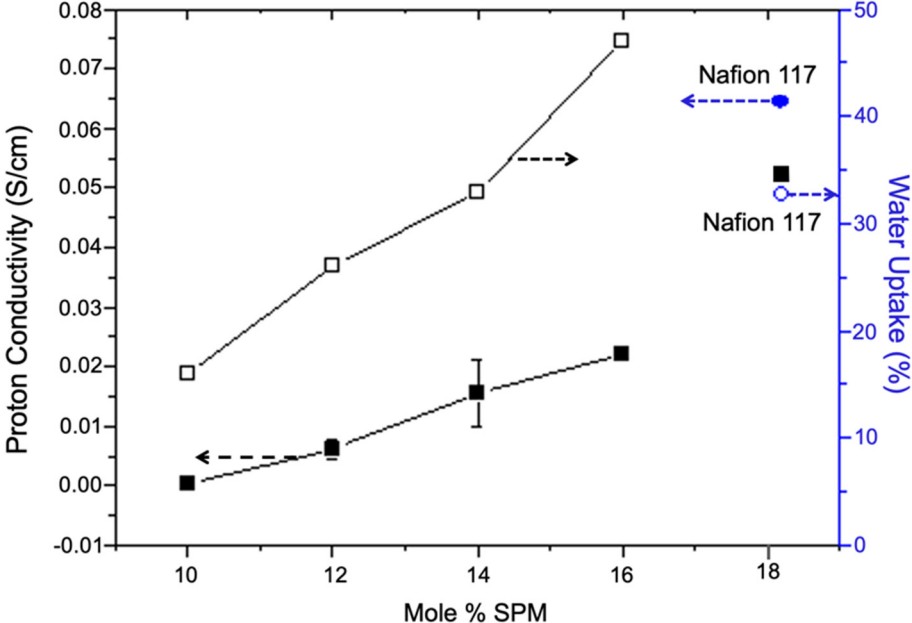

**Figure 11.** Influence of ionic SPM groups on the proton conductivities of the S20-Y membranes at room temperature, and the Nafion® 117 membrane is included for comparison.

Quite fortuitously, the increase in TFPM content also enables more SPM units to be incorporated into the ionomer (Table 1). The distribution of the ionic SPM units in the quaternary ionomer with the mediating role of TFPM impacts proton conductivity. The S20-Y membranes (Y = 10–18 mol% SPM, Figure 11) shows the trend that the increase in SPM content leads to a concomitant increase in water uptake and proton conductivity. Its structural origin lies in the rise of ionic (SPM) content, which results in longer SPM segments well distributed in the ionomer copolymer chains. This is the requirement for constructing interconnecting, dense and sizable proton transport channels. [37] As shown in Figure 11, the S20-18 membrane attains the highest proton conductivity ($\sigma = 0.05$ S/cm),

it is almost an order of magnitude improvement over the proton conductivity of the S0-10 membrane. This agrees well with the above inference on the ionic conducting channels. The presence of 20 mol% of TFPM could permit 18 mol% SPM (in the quaternary copolymer) to be spatially able to form proton conducting channels throughout the membrane matrix. Although it was experimentally possible to increase the SPM content further by using more TFPM in the synthesis, the resulting PEM properties are not what expected. For example, the S25-Y membranes exhibit rather poor proton conductivities, which is attributed to the fact that that once SPM content in S25-Y reaches a certain level, the hydrophobic TFPM segments drive the aggregation of SPM segments (forming hydrophilic domains when the membrane is cast). This would leave behind segregated hydrophilic domains at the cost of diminishing $H^+$ transfer channels. Such a membrane matrix structure, on the other hand, gives rise to excessive water swelling in these hydrophilic domains, which become inferior mechanical spots of the membrane.

## 4. Conclusions

This study focuses on the topic of promoting the ionic component in a type of random aliphatic ionomer via introduction of a fluorine-containing acrylate monomer (TFPM) in the solution copolymerization system. The possible reasons for the increase in the ionic characteristic of the random ionomers were inferred by examining variations of reduced viscosity, zeta potentials, and light scattering intensity with the composition and solution concentration of ionomer. The characterization results were supplemented by spectroscopic (UV-Vis and $^1$H-NMR) and electron microscopic images.

It is suggested that associative interactions via H-bonding between the TFPM and ionic monomer (SPM) assist with distributing SPM more uniformly in the ionomer backbone. As a result, the clustering of SPM segments causing an early phase separation problem in the synthesis of the ternary ionomers is effectively moderated (i.e., the content of SPM in copolymer is lifted from 10 mol% to above 18% in the presence of 20 mol% of TFPM). The TFPM addition also brings about noticeably different solution behaviors from those of the ternary random ionomers, mostly due to the modification of the solvent accessibility of the SPM segments as well as a more uniform distribution of AN segments in the copolymer chain. This study also demonstrates that the hydrophobic TFPM and ionic SPM units affect the structural and functional properties of the membranes in opposite ways, and hence composition optimization is critical for developing workable membranes. The main result of this study is the attainment of a membrane possessing a proton conductivity of 0.05 S/cm. The membrane consists of 20 mol% TFPM and 18 mol% SPM except for the other two structural monomeric units.

**Author Contributions:** Conceptualization, D.J., J.Y.L. and L.H.; methodology, D.J. and C.F.; validation, L.H. and J.Y.L.; formal analysis, L.H.; investigation, D.J. and C.F.; resources, J.Y.L. and L.H.; writing—original draft preparation, D.J.; writing—review and editing, J.Y.L. and L.H.; visualization, L.H.; supervision, J.Y.L. and L.H.; project administration, J.Y.L.; funding acquisition, J.Y.L. All authors have read and agreed to the published version of the manuscript.

**Funding:** This research received no external funding.

**Institutional Review Board Statement:** Not applicable.

**Informed Consent Statement:** Not applicable.

**Data Availability Statement:** Not applicable.

**Acknowledgments:** This investigation was completed at the Department of Chemical & Biomolecular Engineering of National University of Singapore.

**Conflicts of Interest:** The authors declare no conflict of interest.

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
