# Peer review of "Mitigating Early Phase Separation of Aliphatic Random Ionomers by the Hydrophobic H-Bond Acceptor Addition"

_jcs, doi:10.3390/jcs6030073_

Round 1
Reviewer 1 Report
The paper clearly presents a comprehensive study to demonstrate the effect of a hydrophobic modifier. However the following points should be considered for further improvement in the readability of the paper.
- In the abstract you need to connect your study to the real world application and mention the significance of your study
- Remove the numbers from the keywords
- Minor english corrections and formatting changes are required and they can be found in the comments of the attached pdf file
- At the end of your results and discussions section you mentioned few limitations of this study. Please elaborate the points whether you have any results or image to support this. How these limitations undermine the results of this study from practical application point of view.
- Most of the reference are old except couple of them are from 2020 and 2021. Few more latest references need to be reviewed in the introduction section.

Author Response
Reviewer 1:
Comment 1. In the abstract you need to connect your study to the real world application and mention the significance of your study
Response: Thanks, a statement has been added to the abstract, please refer to the last sentence.
Comment 2. Remove the numbers from the keywords
Response: Removed as requested.
Comment 3. Minor english corrections and formatting changes are required and they can be found in the comments of the attached pdf file
Response: Thanks for point out this weak point. All the points annotated in the pdf copy have been amended, and additionally, we have carefully gone through the entire text and made numerous changes to improve the quality of presentation. All these changes are marked in red.
Comment 4. At the end of your results and discussions section you mentioned few limitations of this study. Please elaborate the points whether you have any results or image to support this. How these limitations undermine the results of this study from practical application point of view.
Response: Indeed, the limitation described in the last paragraph of Section 3.4 is about the content of the ionic monomer unit. As requested, we have revised this paragraph to propose the structural origin causing the limitation.
Comment 5. Most of the reference are old except couple of them are from 2020 and 2021. Few more latest references need to be reviewed in the introduction section.
Response: Thanks for pointing this flaw. We have included six more recent publications (4, 6, 8, 11, 17, and 37) in the manuscript.
Reviewer 2 Report
“This report describes” – this is a research paper and not a report !
Please check the entire paper for different typos – spaces between units and values and so on !
The quality of Figure 4 is low please provide a better quality
The conclusions are too verbally they requires endorsement by quantitative data linked to results gathered in this work
Most of references are out of date – new references are required
Author Response
Comment 1. Please check the entire paper for different typos – spaces between units and values and so on !
Response: Thanks for pointing out this flaw. We have gone through the text to correct typos and uniform the spaces in numbers and units.
Comment 2. The quality of Figure 4 is low please provide a better quality
Response: Thanks, both images have been updated.
Comment 3. The conclusions are too verbally they requires endorsement by quantitative data linked to results gathered in this work
Response: Thanks for offering this viewpoint, we have modified the conclusion by including two key quantitative figures.
Comment 4. Most of references are out of date – new references are required
Response: Thanks for pointing this flaw. We have included six more recent publications (4, 6, 8, 11, 17, and 37) in the manuscript.